# Preclinical Marmoset Model for Targeting Chronic Inflammation as a Strategy to Prevent Alzheimer’s Disease

**DOI:** 10.3390/vaccines9040388

**Published:** 2021-04-15

**Authors:** Ingrid H. C. H. M. Philippens, Jan A. M. Langermans

**Affiliations:** 1Animal Science Department, Biomedical Primate Research Centre (BPRC), Lange Kleiweg 161, 2288 GJ Rijswijk, The Netherlands; langermans@bprc.nl; 2Department Population Health Sciences, Unit Animals in Science and Society, Faculty of Veterinary Medicine, Utrecht University, Yalelaan 1, 3584 CL Utrecht, The Netherlands

**Keywords:** Alzheimer’s disease, amyloidosis, amyloid-beta plaques, inflammation, microglia, non-human primates

## Abstract

Due to the aging population, modern society is facing an increasing prevalence of neurological diseases such as Alzheimer’s disease (AD). AD is an age-related chronic neurodegenerative disorder for which no satisfying therapy exists. Understanding the mechanisms underlying the onset of AD is necessary to find targets for protective treatment. There is growing awareness of the essential role of the immune system in the early AD pathology. Amyloidopathy, the main feature of early-stage AD, has a deregulating effect on the immune function. This is reciprocal as the immune system also affects amyloidopathy. It seems that the inflammatory reaction shows a heterogeneous pattern depending on the stage of the disease and the variation between individuals, making not only the target but also the timing of treatment important. The lack of relevant translational animal models that faithfully reproduce clinical and pathogenic features of AD is a major cause of the delay in developing new disease-modifying therapies and their optimal timing of administration. This review describes the communication between amyloidopathy and inflammation and the possibility of using nonhuman primates as a relevant animal model for preclinical AD research.

## 1. Introduction

Alzheimer’s disease (AD) is a serious age-related chronic neurodegenerative disorder that is increasingly common due to an aging population. Today, more than 46 million people worldwide suffer from AD. That number will increase to at least 135 million by 2050 [1]. The problem has been widely recognized and triggered research efforts to better understand the biology of aging and to develop strategies to find treatment targets to limit the progression of AD. While all the research efforts to develop strategies to prevent or slow down the progression of AD, there is still no satisfying therapy for this disease. To identify targets for effective drug therapies, it is of paramount importance to understand the complexity of multi-pathogenic processes. Studying neurobiology in combination with disease manifestation in patients is difficult and is limited to clinical trials and post-mortem research. Animal studies, and the nonhuman primate (NHP) in particular, offer the opportunity to study the role and impact of different processes, such as the role of inflammation and amyloidosis in the early stage of AD. In addition, the marmoset monkey (*Callithrix jacchus*) also offers the opportunity to study age-related cognitive decline [2,3,4].

Besides memory impairment and general cognitive decline, multiple pathogenic determinants have been identified in AD, of which senile plaques (SPs), consisting mainly of β-amyloid (Aβ) aggregates, and intracellular neurofibrillary tau tangles, are the most intensely investigated [5]. However, neurofibrillary tau pathology is not exclusive for AD. Tauopathies occur in many brain disorders [6,7]. Amyloidopathy, on the other hand, is recognized as the earliest feature of AD and forms the basis of the amyloid cascade hypothesis first stated by Hardy et al. [8]. Amyloidopathy results from proteolytic cleavage of the β-amyloid precursor protein (APP) in amyloid protein (Aβ) with a variable length of 40 to 43 amino acids by α-secretase and β-secretase at the N-terminal and γ-secretase at the C-terminal of APP [9] (Figure 1). The outermost extracellular fragment of APP is cleaved by β-secretase and released extracellularly. The remaining part of APP is then cleaved by γ-secretase, releasing extracellular Aβ monomer. Because γ-secretase has variable cleavage sites, the length of Aβ will be between 38 and 43 amino acids. The Aβ_42–43_ monomers are insoluble and will aggregate into toxic dimers, oligomers, and fibrils that will eventually form senile plaques [10,11,12]. The presence of APP C-terminal fragments appears to correlate with the development of neurofilament tau tangles in cells [13]. This raises the possibility that disturbed APP metabolism affects the spread of tau pathologies from cell to cell in AD, and thus abnormalities in APP metabolism may cause aggregation of tau. This is in association with the fact that people with a triplication of chromosome 21, as seen in Down syndrome, overexpress the APP gene that drive AD pathology in these individuals at a young age [14].

In addition to these well-known features of AD, other pathologies, such as inflammation, are increasingly associated with AD [15]. Chronic inflammation like the pro-inflammatory state that arises with aging, also referred to as “inflammaging” [16] is recognized as an important part of the AD pathogenesis and is clearly linked to the amyloidopathy in early-AD, which forms the basis of the amyloid cascade hypothesis [15,17,18,19,20,21,22,23,24,25]. Indeed, the onset of the amyloidopathy is associated with immune activation, as the early intracellular accumulation of Aβ oligomers elicits an inflammatory response [21,25,26,27]. In addition, expression of CD95 (apoptosis receptor) and CD45RA (maturity marker) on CD4+ T-cell reflect immune activation in the blood and are predictive for AD, indicating immune interference [28,29,30]. However, the causality of Aβ accumulation and activation of the immune system is a matter of debate, as chronic inflammation is measured prior to the onset of AD and could induce Aβ accumulation by reactive microglial cells and pro-inflammatory cytokine secretion [23,31,32]. Because this relationship is difficult to investigate in humans, animal models can be an important link in learning more about the cause and effects. There is a differential relationship between the glial cells and the Aβ plaques, as microglia are the driving force in the severe immune reaction associated with Aβ and neurodegeneration in general, whereas the exact role of astrocytes remains unclear [33,34,35,36,37]. This review discusses some of the cellular players in the immune-related aspects of AD in relation to amyloidopathy and how the marmoset monkey can be used as a model to learn more about the role of the immune system in AD.

## 2. Astrocytes and Microglia

It was believed that astrocytes actively participate in Aβ clearance [33], but stimulation of tumor necrosis factor (TNF)-α and interferon (IFN)-γ in combination with Aβ have been shown to result in the production of Aβ in astrocytes. This makes astrocytes a large source of Aβ in AD, as they greatly outnumber the amount of neurons [38]. On the other hand, astrocytes actively participate in limiting plaque growth in APP/PS1 mice [39]. In addition to a detrimental or protective effect of astrocytes on Aβ, they can also be damaged by the presence of Aβ [40]. Aβ has a more serious deleterious effect on astrocytes than on neurons due to an increase of intracellular Ca^2+^ levels followed by an increase of reactive oxygen species (ROS), resulting in impaired astrocyte functioning [40]. Although the exact functionality of astrocytes in AD pathology remains unclear, the long-term exposure of astrocytes to increasing levels of Aβ will ultimately result in a reduced protective effect of the astrocyte in AD [41]. In contrast, the role of microglia is widely recognized as a prominent feature in AD [42]. Microglia have a variety of activated phenotypes: proinflammation (M1), allergic response (M2a), tissue repair (M2c), and anti-inflammation phenotypes (M2c activation) [43]. Upon activation, they change the morphology from a ramified shape into an amoeboid shape (Figure 2) and proliferate [36,44].

The type of activation depends on its stimulation, as the M1 response is the stereotype pro-inflammatory and phagocytic reaction on an acute infection and IFNγ, TNFα, or lipopolysaccharide (LPS) stimulation, whereas interleukin (IL) 10 stimulation of microglia results in M2c activation, which dampens the immune response [43]. Microglia bind to Aβ via receptors including class A scavenger receptor A1, CD36, CD14, α6β1 integrin, CD47 and toll like receptors (TLR2, TLR4, TLR6, and TLR9) (Figure 2), which are believed to be part of the inflammatory reaction in AD [20]. The binding of Aβ with CD36, TLR4 and TLR6 results in activation of microglia that start to produce proinflammatory cytokines and chemokines [20]. Early-AD is usually linked to M1 activation, as the Aβ proteins activate the microglia cells by binding pattern recognition receptors (PRRs), such as complement, Fc receptors, CD33, triggering receptor expressed on myeloid cells 2 (TREM2), receptor for advanced glycation endproducts (RAGE), scavenger and TLR receptors, and in this way triggering NF-κB translocation and thus an enhancement of the immune response [24,45,46,47]. Although the receptors initiate a pro-inflammatory response and a neurotoxic respiratory burst upon Aβ binding, each receptor has a specific mechanism. Next to this, the formation of the Aβ protein in the soluble amyloid-beta (AβO) or fibrillar amyloid-beta (AβF) also results in divergent microglial activation [48,49]. AβO binds the Ca^2+^ activated potassium channel kCa3.1 resulting in specific NO generation [49], whereas AβF binding to the CD36 receptor leads to internalization of the Aβ protein, activation of nucleotide-binding oligomerization domain (NOD)-like receptor pyrin protein 3 (NLRP3) receptor in the lysosome, and subsequently the enzymatic transformation of pro-IL1β to the pro-inflammatory cytokine IL1β [47]. In response to receptor ligation, microglia begin to phagocytose Aβ fibrils. As a result, these fibrils enter the endosomal/lysosomal pathway. In contrast to fibrillar Aβ, which is largely resistant to enzymatic degradation, soluble Aβ can be degraded by a variety of extracellular proteases [20]. RAGE receptor binding [33,50,51,52], on the other hand, results in the secretion of macrophage colony-stimulating factor (m-CSF), which recruits more microglia [15,46], whereas scavenger RA binding result in ROS secretion via activation of the nicotinamide adenine dinucleotide phosphate (NADPH) oxidase complex exerting neurotoxic effects [15,17,49,53].

Besides activation of microglia through the interaction with Aβ protein, neuronal debris released into the extracellular space after neuronal death such as laminin, MMP3, or neuromelanin also results in microglial M1 activation [54]. Besides the prominent role of M1 microglia in AD, the immune response in early-AD is heterogeneous, as in some individuals not only M1, but also the M2a microglial phenotype is recognized, suggesting a comorbidity [55]. The M2a phenotype is described to be an allergic response, but it is also involved in Aβ agglomeration [55]. On the other hand, aged microglia themselves show increased synthesis of lipid messenger prostaglandin E_2_ (PGE_2_), an important modulator of inflammation [56]. This upregulation of PGE_2_ has been shown to enhance the processes seen in AD, such as cognitive decline in aged mice [56]. Increased levels of PGE_2_ are also found in ageing and in neurodegenerative diseases [57]. These findings indicate a complex heterogeneous immune response that is dependent on the stage of AD.

## 3. Inflammaging

As described above, immune activation balances between pro-(M1) and anti-(M2) inflammation, which can temporally be tilted towards pro-inflammation after an acute infection and later counterbalanced with anti-inflammation in a healthy reaction. During aging, this equilibrium can be deregulated and chronically tilted towards pro-inflammation, also called inflammaging [16]. Earlier encounters that primed immune cells, chronic systemic inflammation or genetic predisposition are explanations for the inflammaging phenomenon [58,59,60,61]. In persons that demonstrate extreme longevity, the chronic pro-inflammation is counterbalanced by strong anti-inflammation mechanisms to eliminate the toxic side effect of pro-inflammation [16]. Then again, inflammaging is increasingly mentioned as a risk factor for neurodegenerative diseases [16,17,30,31,62,63]. Inflammaging consists of specifically upregulation of the innate mononuclear immunity, like macrophages and microglia, and Th1 cells resulting in pro-inflammatory cytokine secretion (IFNγ, IL1β, IL6, and TNFα), whereas the adaptive immunity is downregulated [17,62,63]. The aging effect on microglia includes subtle changes in morphology [59,64] and a higher susceptibility for M1 over-reactivity accompanied by a more pronounced pro-inflammatory cytokine expression profile [44,65,66]. In addition to over-active microglia, replicative senescent and dystrophic microglia are also present in the aging brain [66,67]. Nevertheless, the tissue damage due to the toxic age-related proinflammatory state may participate in the onset of Aβ agglomeration, which is consistent with the increased incidence of AD in patients with increased cytokines due to polymorphisms in genes of plaque associated cytokines, namely TNF-α, IL-1, IL-6, and acute phase proteins (α1-antichymotrypsin) [31,68,69,70]. The increase in susceptibility to AD by inflammaging can be explained in part by the direct effect of IL-1β on amyloidopathy. The high levels of IL-1β result in an increase in the expression of neuronal APP [31,45,71,72] and its processing in addition to the activation of astrocyte, leading to overexpression of S100B, which also induces APP synthesis [73]. Additionally, proinflammatory M1 phenotypic microglia reaction via LPS injection directly resulted in amyloidosis, indicating an Aβ-promoting effect of the M1 inflammation [32]. Taken together, it seems evident that M1 activation of microglia in combination with proinflammatory cytokine secretion (IL-1β, IL-6, TNF-α, INF-γ) and the associated reactive oxygen/nitrogen species (ROS/RNS) secretion result in an increase in AD susceptibility via indirect tissue damage or direct enhancement of Aβ synthesis and agglomeration [31,35,46,54,74,75,76,77].

## 4. Difficulties in Finding Targets for Treatment

In literature, the dual role of the immune system is mentioned, as microglial neuroprotection would occur in early-AD by Aβ protein phagocytosis, while the increase of pro-inflammatory cytokines would later result in neurotoxicity [59,78]. Several publications have been devoted to this dual role of the immune system described as a double-edged sword [79]. The neuroprotective properties of controlled immune activation have been mentioned before, although there was no doubt about the neurotoxic effects of an exaggerated immune response [80]. They specifically attribute the neuroprotective effects to infiltrating bone marrow derived microglia (BMDM) that would be more effective antigen presenting cells than the resident self-reproducing microglial cells in the brain, as the latter have low levels of major histocompatibility complex (MHC) class II receptors and CD11c [80,81,82]. The BMDMs are said to migrate specifically to brain regions that exhibit neurodegeneration, but recruitment of these cells is a slow process in which manipulation could prove beneficial [80,81]. Although recruitment of these BMDMs is one of the options for clearing the amyloid plaques through the use of the immune system, the mechanism and infiltration of BMDMs into the CNS remain a matter of debate [81].

Another way to reduce M1 phenotypic immune activation can be achieved with non-steroidal anti-inflammatory drugs (NSAIDs). NSAIDs target the prostaglandin pathway. Prostaglandins are metabolic products of cyclo-oxygenase (COX) enzymes that become activated upon Aβ stimulation of neurons or immune cells and are involved in a vicious circle between neurons and microglial cells. COX-2 in glial cells result in NF-κB transcription, which in turn results in IL1β secretion that upregulates COX-2 in neurons, leading to PGE_2_ production and subsequently increasing Aβ synthesis [83]. Although targeting the prostaglandin pathway seems a good approach, the results of NSAID studies are controversial. Indeed, rheumatoid arthritis patients treated with NSAIDs showed a lower susceptibility to AD [84]. However, many clinical studies that followed, such as the population-based Rotterdam Study of 306 subjects over a 10-year period, were unable to replicate the proposed association between NSAID use and AD risk [85,86]. However, another study demonstrated the beneficial effect of NSAID treatment in reducing induction of AD before any symptoms appear, making proper timing important for NSAID treatment [87]. The poor results in the NSAID studies could be due to the heterogeneity of immune responses and profiles per patient, as some show a clear M1 expression profile, while others showed an M2a expression profile in early-AD [55]. Unlike NSAID, direct inhibition of PGE_2_ may represent a better approach with a greater specificity targeting not only Aβ synthesis but also restoring cognition [56]. In addition, immune profiles change during the disease course [88], which means that the immune treatment should be adjusted to the immune profile of the patient, making the timing of NSAID treatment important. As AD is a heterogeneous disease and studies in which immunosuppressive treatments have been tested to prevent AD have yielded conflicting results, it is recommended to develop a personalized treatment that can be tailored to the individual’s status. This is especially important for the inflammation-related aspects of AD [43]. This heterogeneity makes it difficult to develop treatment strategies.

## 5. Research to Find Targets for Treatment: Relevance of Marmoset Models

Owing to the complexity of the multi-pathological process and the heterogeneity of the processes during the progression of AD, it will be difficult to develop a well-defined experimental platform for target validation. Studying neurobiology in combination with disease manifestation in patients is difficult. Research in humans is limited to clinical trials and post-mortem material. Animal models offer an opportunity to link neurodegeneration to disease manifestation and an opportunity to explore for intervention therapy strategies (Figure 3).

Currently, no animal models exist that recapitulate all behavioral, physiological, and biochemical aspects of AD. This makes improvement of AD animal models important [89,90,91]. Transgenic mouse models, with mutations in genes involved in APP processing, have been developed that display many of the neuropathological features [90,92]. These transgenic models display a hyper acute model with massive plaque depositions and a different distribution and chemical composition of the plaques compared to humans. This can result in different target sites of the Aβ protein and therefore difficult to translate to the clinic. Furthermore, the immune system of mice differ from the human immune system, which should be considered when using mice for preclinical research [93]. NHPs, on the other hand, have a high resemblance to the immune system and the aging phenotype of humans [94,95] and naturally generate human-sequence Aβ that, with age, aggregates into extracellular senile plaques [96,97,98,99]. The amount of Aβ in the temporal and occipital cortices of aged macaques reaches levels comparable to cortical Aβ levels in Alzheimer’s patients [100]. It has recently been established that aging rhesus macaques also exhibit tau pathology in the same qualitative pattern and sequence as seen in humans with AD [101]. The same observation was made in the marmoset monkey; spontaneous age-related Aβ deposition and aggregation of abnormally phosphorylated tau protein were found in old marmosets [102] similar to what is found in humans [103]. In contrast, it was also recently reported that tau isoforms and their phosphorylation status in the brain of healthy marmosets differ from human tau isoforms [104]. However, these studies have been conducted in young animals. Although both rhesus macaques and marmosets are useful models for neurological disorders, including AD [105,106], the marmoset has several advantages due to its small size, relatively short reproduction time, birth of twins, shorter life cycle compared to rhesus monkeys, and the availability of transgenic models [107].

Intracerebral injections of Aβ containing homogenates in marmoset monkeys activate plaque progression, even in young individuals [108,109]. Foreign tissue, such as brain homogenate, can stimulate the local immune system to a certain level, which could be sufficient for this activation of plaque progression. Injection of pure synthetic Aβ-peptides did not elicit plaque formation [109]. This suggests that a pro-inflammatory cytokine profile, activated by inflammation, combined with diffuse amyloid depositions, may initiate a self-propagating process leading to plaque progression and thus the risk of developing AD [106,110]. Activation of the immune system by intracranial LPS injection in combination with insoluble fibrillar Aβ_42–43_ indeed accelerates amyloidosis in marmoset monkeys [106] (Figure 4), which is in line with the finding that inflammatory mediators activate Aβ aggregation [110].

In Figure 5, an example is shown of the development of plaques in the presence of activated microglia in the brain of a marmoset monkey 5 months after an intracranial injection of LPS and fibrillar Aβ_43_ (Bachem H-1586, Bachem AG, Bubendorf, Switzerland). The monkeys that were injected with only Aβ_43_ did not develop these amyloid plaques [106]. The role of the immune system is highlighted by the fact that natural amyloidosis was found in brain tissue from a middle-aged marmoset (6 years) that had succumbed from wasting syndrome, a clinical condition associated with inflammatory bowel disease (chronic colitis), a chronic systemic inflammation [111,112]. Additionally, in the marmoset model for amyloidopathy, an upregulation of the maturity marker CD45RA and a downregulation of the apoptosis marker CD95 on CD4+ T-cells was observed, reflecting peripheral immune activation as is seen in AD patients [28,29,30].

These findings offer opportunities to study the processes of the interaction between the immune system and plaque progression in animal models, in which this process can be stimulated. NHPs have been mentioned as a model for Alzheimer’s-related research and drug discovery [113]. These animals are closely related to humans and recapitulate the human aging process and development of age-related diseases [114,115]. The close evolutionary consistency including the genome similarity and both the face and construct validity of the human aging process and development of age-related diseases, making NHP an ideal translational model. Several relevant aspects between rodents, marmosets, and humans are listed in Table 1. During the 65 million years of evolutionary separation between rodents and primates, neuronal pathways and cognitive capacities have evolved in distinct directions, resulting in specialized brain structures for perceptual and cognitive capacities in primates, including humans [116]. Similarities between human and primates include encephalization and sulcal characteristics, similar numbers and densities of cortical neurons, a large prefrontal cortex containing areas responsible for working memory, executive function, and aspects of decision-making, similar nuclear organization, projection pathways, and innervation pathways of the hippocampus, analogous blood–brain barrier structure and functioning, and the existence of mirror neurons [113]. NHP also exhibit a longer average lifespan compared to rodents and exhibit naturally occurring age-associated diseases with a developmental course that closely resembles human conditions [117]. Indeed, comprehensive studies have shown that data obtained from NHP are the most predictive for human pharmacokinetic parameters [118]. More specifically, the marmoset appears to be a good preclinical translational model for immune-related diseases and neuroscience [119]. It also appears that, in marmosets, microglia activation persists long after injury and the progression of the microglia response persists, while, in rodents, it appears in peaks and disappears rapidly after injury [120]. Unlike rodents, marmosets also have a similar analog of the major histocompatibility complex (MHC) class II in humans, the Caja-DR and DQ (from *Callithrix jacchus*) [121], which is associated with activated microglia with AD lesions in humans [122]. The common marmoset, a small Neotropical New World primate, holds particular promise as a model for aging diseases [114]. Common marmosets share ~93% sequence identity with the human genome [123,124]. This also results in a close resemblance to the immune system and aging phenotype of humans [94,95], making the common marmoset an appealing aging model [95,125,126] and especially a useful (sporadic) AD animal model [98,108,127]. Key components of the antibody response are functionally conserved between lower primates and humans [128]. The common marmoset may be useful as an in vivo model of immune function, particularly with regard to the role of interleukins [128]. Although the identity percentage of the immunome proteins with humans is very high in rhesus monkeys (96.77%), these proteins still have a strong overlap in evolutionarily more distant marmosets with humans of 94.11% [129]. In addition, the immunological changes in common marmosets over their life span revealed several similarities to age-related changes in humans [130]. Marmosets are outbred species raised under normal exposure to antigens, which have allowed their immune systems to develop normally as is the case in humans. This leads to more variation, but also to a more faithful representation of the responses to injuries. Marmosets have the advantage of giving birth to twins that share the same placenta and are therefore hematopoietic chimeras, resulting in a similar innate immune system. For studies of the interaction between the immune system and AD progression, these chimeric twins limit variability across different treatment groups [131]. The potential for the development of transgenic common marmosets [107] opens new ways to identify pathways involved in the development of neuropathies and to study the role of the inflammatory responses [132].

Marmosets have proven to be valuable models to study inflammatory pathways and to analyze new therapies and drugs in another neurodegenerative disease such as Parkinson’s disease [37,133,134]. Not only with regard to neuropathies, such as alpha-synuclein [135], but also for the functional motor aspects [136,137], including dyskinesia [138], and the non-motor aspects such as sleep disturbances that also plays an important role in Alzheimer’s disease [139]. Marmosets are also essential to study the early aspects of such diseases, which is not feasible in humans. Examples are the validation of disease-modifying therapies [140], research towards susceptibility and the dynamics of the progression of Parkinson’s disease [141,142], and the involvement of compensatory mechanisms [143]. Although the use of NHP raises ethical hurdles that could limit its application to preclinical studies, the small marmoset (300–450 g) is an attractive model for translational research. The marmoset model covers most of the basic pathogenic mechanisms of the human disease, including protein misfolding and inflammation. Furthermore, only small quantities of candidate drugs are required in this small NHP model.

## 6. Conclusions

AD is a devastating neurodegenerative disorder with an increasing occurrence due to a growing population that affects not only the patients, but their families and friends as well. To understand and prevent this disease, markers are needed that predict and are involved in the onset of AD. Aging remains the greatest risk factor for AD. During aging, challenges and experiences shape, mature, and prime the human body, including its immune system [144]. In most individuals, this leads to a low-grade chronic inflammatory state called inflammaging, which makes them more prone to overreaction after stimulation. Indeed, there is growing awareness of the essential role of the immune system in the early stage of AD pathology, making this bodily mechanism relevant for research in relation to AD.

Trials in which they tested immunosuppression to test its preventative effect on the onset of AD led to conflicting results. Since AD is a dynamic heterogeneous disease, personalized treatment that can be adjusted to the status of the individual in need for heterogeneous treatment, especially for the inflammation associated with AD, should be considered [43]. However, before doing so, we need to know more about the influence of the immune system on AD progression. For this, we need research models that cover both the immune and amyloidopathy aspects of AD. The anatomical, immunological, and functional organization in the brain, and the fact that marmoset monkeys display spontaneous development of amyloid plaques, makes this animal species valuable for translational studies within AD research.

## Figures and Tables

**Figure 1 vaccines-09-00388-f001:**
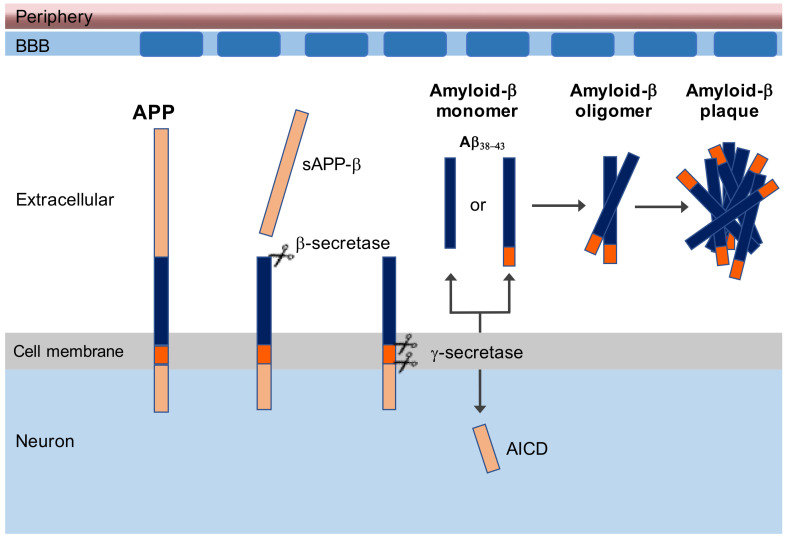
Amyloid-beta plaque generated from amyloid precursor protein (APP). Firstly, the outermost extracellular fragment of APP is cleaved by β-secretase, and sAPP-β is released extracellularly. The remaining part of APP is subsequently cleaved by γ-secretase, freeing extracellular amyloid-β (Aβ) monomer and an intracellular APP domain (AICD). Because γ-secretase has variable cleavage sites, the length of Aβ will be between 38 and 43 amino acids long. The Aβ can aggregate into oligomers, and the Aβ_42–43_ will form plaques.

**Figure 2 vaccines-09-00388-f002:**
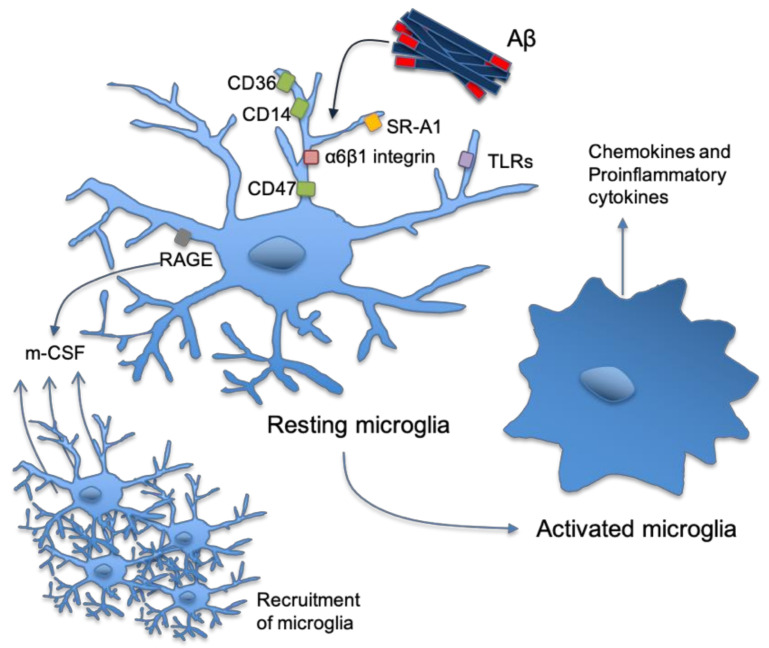
Microglia-amyloid-β interaction. Microglia bind to Aβ via receptors including class A scavenger receptor A1, CD36, CD14, α6β1 integrin, CD47, and toll like receptors (TLR2, TLR4, TLR6, and TLR9). The binding of Aβ with CD36, TLR4, and TLR6 results in activation of microglia which start to produce proinflammatory cytokines and chemokines. Binding to RAGE receptor results in the secretion of m-CSF, which recruits more microglia, whereas scavenger RA binding results in ROS secretion via activation of the NADPH oxidase complex exerting neurotoxic effects.

**Figure 3 vaccines-09-00388-f003:**
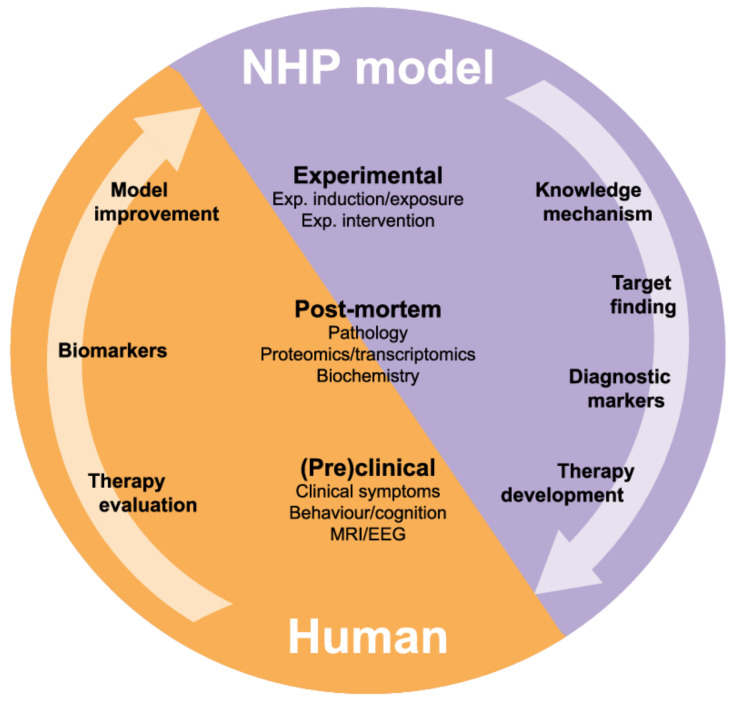
Translational model for mechanistic research and development of therapeutic interventions. Overlapping (pre)clinical and post-mortem markers between human and nonhuman primate (NHP) models increase the extrapolation of the results. The correlation between the experimental (invasive) interventions and the invasive/post-mortem parameters from NHP studies will provide relevant knowledge about the underlying mechanism and will help to find new targets for treatment.

**Figure 4 vaccines-09-00388-f004:**
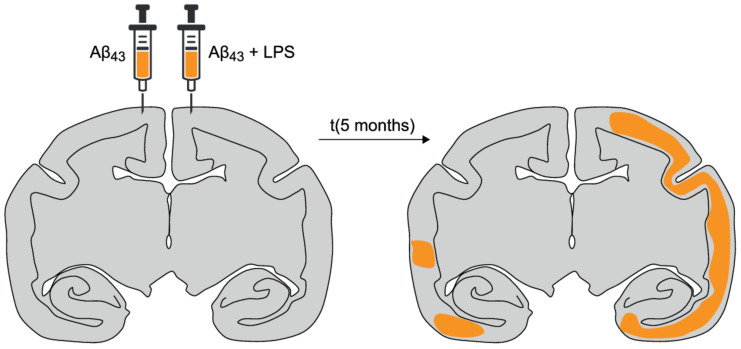
Plaque load and distribution in the brain of a marmoset monkey. The distribution of Aβ plaques measured five months after intracranial injection of Aβ with or without LPS (left) is visualized on a transcranial section (AP-0) and indicated in orange (right). The hemisphere in which Aβ_43_ + LPS was injected showed significantly more severe amyloidopathy than the other hemisphere (*p* < 0.001) [106]. Although the injection was given in the cortex 4 mm from the sutura sagittalis, the Aβ plaques were distributed over the temporal and posterior cortex.

**Figure 5 vaccines-09-00388-f005:**
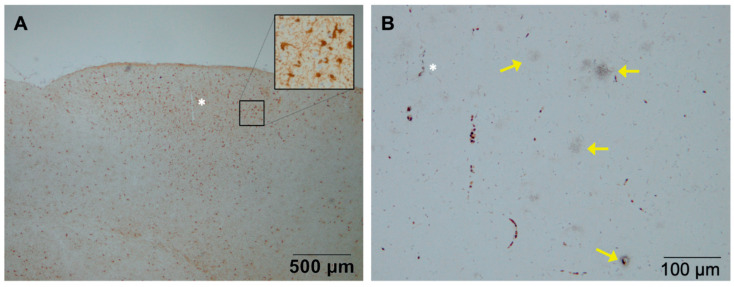
Reactive microglia surrounding senile plaque densities in a marmoset monkey (source Philippens, BPRC). Regions with activated microglia accumulation were visible (**A**) (Iba1 staining). Diffuse senile plaque formation co-localized with the reactive microglia as visible in a mirror section (**B**) (Campbell–Switzer staining). The vessel indicated by the asterix (*) symbol, was used for navigation between (**A**) and (**B**).

**Table 1 vaccines-09-00388-t001:** Different aspects between rodents (mice), marmoset, and human.

Aspect	Mice	Marmoset	Human
Evolutionary distance (million years)	65	35	0
Life expectancy (years)	2	10–15	80
Reproduction (number per year)	50	2–4	1
Genetic diversity	inbred	outbred	outbred
Environment	standardized	standardized	variable
Main sense	smell	vision	vision
Cortical thickness (mm)	1.21	2.37	2.62
Sleep pattern	nocturnal	diurnal	diurnal

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
