# Peer review of "Preclinical Marmoset Model for Targeting Chronic Inflammation as a Strategy to Prevent Alzheimer’s Disease"

_vaccines, 2021, doi:10.3390/vaccines9040388_

Round 1
Reviewer 1 Report
In this manuscript Authors propose the use of non-human primates (NHPs) to model from experimental point of view the complexity of pathophysiology of Alzheimer’s disease (AD) and anti- AD drug evaluation. Specifically, Authors propose to use marmosets (small monkeys) who show spontaneous development of amyloid plaques, makes this animal species valuable for translational studies within AD research. Authors pointed out that marmosets show high resemblance of the human immune system and aging phenotype thus representing an appealing model of sporadic AD.
The manuscript covers in a comprehensive way the current view of the role of the immune system in the development of AD. I have a few comments to offer to the Authors’ attention.
- Authors stated that NHPs may represent a model for AD drug discovery and cite an academic paper to support this statement (Van Dam et al. Non human primate models for Alzheimer's disease-related research and drug discovery. Expert Opin Drug Discov 2017, 12, 187-200). A part the obvious ethical considerations on the use of NHPs in AD drug discovery, there are logistic and feasibility problems to consider when speaking of clinical candidate selection within a modern drug discovery process, especially at the pharmaceutical companies.
- In the section describing the dysfunction of the immune system in the development of AD, Authors may cite and comment the recent article published on Nature of the role of metabolic dysfunction of myeloid PGE2 in cognitive decline and the possibility to reverse these aging effects by selective blocking the EP2 receptor subtype of PGE2 (Minhas et al. Restoring metabolism of myeloid cells reverses cognitive decline in ageing. Nature 2021; 590: 122-128).
- Authors may comment more explicitly on the hypothesis that anti-inflammatory clinical trials in AD failed because they included patients with both normal, exaggerated or impaired central and peripheral inflammatory status. Thus, potential beneficial effects of anti-inflammatory drugs on patients with exaggerated inflammatory tone were obscured by the detrimental effects on patients with attenuated inflammatory tone. So, only patients with over-reacting inflammatory system should be enrolled in anti-inflammatory clinical trials and this could be reached by appropriate immune status assessment (human monocyte-derived macrophages ?) at screening according to a precision medicine approach.
- As minor points, in the Introduction, I suggest providing a reference to support present and future prevalence of AD. In addition, please note that the metabolism of APP to generate Aβ does not involve α-secretase.
Author Response
We are grateful for the expeditious review of our manuscript entitled “ Preclinical marmoset model for targeting chronic inflammation as a strategy to prevent Alzheimer’s disease” by Philippens and Langermans. This review is focused on the use of the marmoset as a model for drug development against Alzheimer’s disease and is written upon invitation for the special issue nonhuman primates in immune modulation and drug discovery. We sincerely appreciate the positive feedback, thoughtful comments and helpful suggestions for the improvement of the paper.
We have now addressed each issue carefully and incorporated the suggestions in our revised manuscript. The specific changes made in the manuscript are listed in greater detail below:
1) At the end of the manuscript under section 5 we added a remark about ethical hurdles and feasibility problems.
Line 344-349: Although the use of NHP evokes ethical hurdles that might limit the application for preclinical studies, the small marmoset (300-450 grams) is an attractive model for translational research. The marmoset model covers most of the basic pathogenic mechanisms of the human disease, including protein misfolding and inflammation. Furthermore, only small amounts of candidate drugs is needed in this small NHP model.
2) The reference Minhas et al. 2021 has been added in the text at the end of section 2 and under section 4.
Section 2, line 122-127: On the other hand, aged microglia themselves show increased synthesis of lipid messenger prostaglandin E2(PGE2), which is an important modulator of inflammation [56]. This upregulation of PGE2has been shown to enhance the processes seen in AD, such as cognitive decline in eldery mice [56]. Increased levels of PGE2are also found in ageing and in neurodegenerative diseases [57].
Section 4, line 208-209: Unlike NSAID, inhibition of PGE2may represent a better approach with a greater specificity that not only target the Aβ synthesis but also restore cognition [56].
3) A remark about personalized treatment in AD is added under section 4.
Line 212-216: As AD is a heterogeneous disease and since studies in which immunosuppressive treatments have been tested for prevention of AD have yielded conflicting results, it is recommended to develop a personalized treatment that can be tailored to the individual's status. This is especially important for the inflammation-related aspects of AD [43].
4) In the introduction (line 32), a reference is added to support the present and future prevalence of AD.
The metabolism of APP to generate amyloid-beta has been clarified in the introduction.
Line 54-57: The outermost extracellular fragment of APP is cleaved by β-secretase and released extracellularly. The remaining part of APP is subsequently cleaved by γ -secretase, freeing extracellular Aβ monomer. Because γ-secretase has variable cleavage sites, the length of Aβ will be between 37 and 43 amino acids long.
Reviewer 2 Report
In this review article, Drs Philippens and Langermans summarized the preclinical application of non-human primate model for targeting chronic inflammation as a strategy to prevent Alzheimer’s disease (AD). They described the communication between amyloidopathy and inflammation and discussed the possibility of using marmoset as relevant animal model for preclinical AD research. As no animal model can faithfully mimic clinical and pathogenic aspects of AD, non-human primates model may help better understand of the pathogenesis and speed the drug development for this disease. Although the whole article is well written, some weaknesses below made this article is not acceptable for publication under current conditions.
- The tile of this review indicates the focus of this article is on the translational potential of non-human primate model in mimicking the interaction between amyloidopathy and inflammation. However, in most of the main text, the authors are overviewing these two pathological events in human patients. More information about the features of amyloidopathy and inflammation in marmoset model of AD should be provided and the advantages/disadvantages of the model in mimicking AD neuropathology should be discussed.
- Some aspects of disease model should be considered when a new disease model is evaluated, such as the appropriateness as analog, transferability of information, genetic uniformity of species, ethical issue, et al.
- Current application of this models in AD mechanism and preclinical test should be summarized.
Author Response
Dear reviewer 2, Manuscript ID: vaccines-1115914
We are grateful for the expeditious review of our manuscript entitled “ Preclinical marmoset model for targeting chronic inflammation as a strategy to prevent Alzheimer’s disease” by Philippens and Langermans. This review is focused on the use of the marmoset as a model for drug development against Alzheimer’s disease (AD) and is written upon invitation for the special issue nonhuman primates (NHP) in immune modulation and drug discovery. We sincerely appreciate the positive feedback, thoughtful comments and helpful suggestions for the improvement of the paper.
We have now addressed each issue carefully and incorporated the suggestions in our revised manuscript. The specific changes made in the manuscript are listed in greater detail below:
There is not yet much information available about the pros and cons of the marmoset model for AD. Therefore, we pointed towards the possibilities to develop a NHP model that mimic most of the features of AD in a natural way. As this is a new model a list about the current applications is too preliminary at this moment. At the end of the manuscript under section 5 we added information about the genetic aspects and the advantages and concerns of the use of NHP.
Line 297-306: During the 65 million years of evolutionary separation between rodents and primates, neuronal pathways and cognitive capacities have evolved in distinct directions, resulted in specialized brain structures for perceptual and cognitive capacities in primates, including human [114]. Similarities between human and primates include encephalization and sulcal characteristics, comparable numbers and densities of cortical neurons, a large prefrontal cortex containing areas that are responsible for working memory, executive function and aspects of decision-making, similar nuclear organization, projection pathways and innervation pathways of the hippocampus, analogous blood–brain barrier structure and functioning, and the existence of mirror neurons [111].
Line 310-316: More specifically, the marmoset appears to be a good preclinical translational model for immune-related diseases and neuroscience [117]. It also appears that in marmosets, microglia activation persists long after injury and the progression of the microglia response persists, while in rodents it appears in peaks and disappears rapidly after injury [118]. In contrast to rodents, marmosets do also have an comparable analogue of the major histocompatibility complex (MHC) class II in human, the Caja-DR and DQ (from Callithrix jacchus)[119], which is associated with activated microglia with AD lesions in human [120].
Line 318-319: Common marmosets share ~93% sequence identity with the human genome [121,122].
Line 325-328: Marmosets are outbred species raised under normal exposure to antigens, which has allowed their immune systems to develop normally as is the case in humans. This leads to more variation, but also to a more true-to-life representation of the responses to injury.
Line 345-349: Although the use of NHP evokes ethical hurdles that might limit the application for preclinical studies, the small marmoset (300-450 grams) is an attractive model for trans-lational research. The marmoset model covers most of the basic pathogenic mechanisms of the human disease, including protein misfolding and inflammation. Furthermore, only small amounts of candidate drugs is needed in this small NHP model.However, substantial ethical hurdles that has to be taken considerably limit their application for preclinical studies. Considering these ethical issues, the small marmoset (300-450 grams) is an attractive model for translational preclinical research as only small amounts of a candidate drugs is needed. Furthermore, the marmoset model covers most of the basic pathogenic mechanisms of the human disease, including protein misfolding and inflammation.
Reviewer 3 Report
The authors have done a review on using marmoset as a preclinical model for targeting chronic inflammation to preventing Alzheimer’s disease. The paper cannot be published in the current form because it does not capture all the studies necessary for a review. Reviews are supposed to be extensive that includes all the important studies that are published, so that the readers could have a complete understanding on the topic.
- The topic in inflammation in Alzheimer’s disease is not adequate. The authors have focused only on astrocytes on microglia, though they are important in inflammation, those are not the only cells responsible for inflammation.
- The authors have given only a paragraph on astrocytes on microglia, which does not include majority of the study that has been published
- The mechanism of inflammation is not detailed although few have been mentioned. A review is supposed to contain a detailed mechanism of the pathways involved with adequate illustrations.
- The pathology of amyloid beta plaques is no where seen in the review, neither are the mechanism of formation.
- The role of immune response which is the basis of the paper is not there in the paper.
- The topic of the paper is the use of marmoset as the preclinical model, but the review does not capture this area at all. Most of it under the preclinical model deals with how monkeys are used as the model but says nothing about marmoset. The topic of the review is no where to be seen.
- The authors must give importance the fonts in the review. Why one of the heading is completely in italics is not been explained. Are they trying to highlight something or was it just a mistake on their part not to pay attention to the final structure?
- Most importantly there is no flow in the review. The authors should take care on how they present their final version. As of now it seems that they have pieced the information together. There is not connection between one and another sentence in many places.
Author Response
Dear reviewer 3, Manuscript ID: vaccines-1115914
We are grateful for the expeditious review of our manuscript entitled “ Preclinical marmoset model for targeting chronic inflammation as a strategy to prevent Alzheimer’s disease” by Philippens and Langermans. This review is focused on the use of the marmoset as a model for drug development against Alzheimer’s disease (AD) and is written upon invitation for the special issue nonhuman primates in immune modulation and drug discovery. We sincerely appreciate the positive feedback, thoughtful comments and helpful suggestions for the improvement of the paper.
We have now addressed each issue carefully and incorporated the suggestions in our revised manuscript. The specific changes made in the manuscript are listed in greater detail below:
1-5) This review is focused on the use of the marmoset as a model for drug development against AD. This is written for the special issue nonhuman primates in immune modulation and drug discovery. Therefore, we did not prepare a review about all immune related aspects seen in AD or all possible mechanisms of the pathways involved, but only focused on these aspects that are relevant for AD as well as for the marmoset model. But we are aware that the introduction was a bit confusing that makes it difficult for the reader to follow the focus of this review. Therefore, we added in the introduction a paragraph in which the question of this review is clarified.
Line 38-43: …it is paramount to have insight into the “complexity of multi-pathogenic processes. Studying neurobiology in combination with disease manifestation in patients is difficult and limited to clinical trials and post-mortem research. Animal studies, and the nonhuman primate (NHP) in particular, offer the opportunity to study the role and impact of different processes, such as the role of inflammation and amyloidosis in the early stage of AD. Furthermore, the marmoset monkey (Callithrix jacchus) gives also the opportunity to study age-related cognitive decline[2-4].
Comment 4, to explain the mechanism of formation of the amyloid plaques a sentence is added in the introduction.
Line 54-57: The outermost extracellular fragment of APP is cleaved by β-secretase and released extracellularly. The remaining part of APP is subsequently cleaved by γ -secretase, freeing extracellular Aβ monomer. Because γ-secretase has a variable cleavage sites, the length of Aβ will be between 37 and 43 amino acids long.
6) The reviewer was wright about the scarce information about the marmoset monkey. We have added several paragraphs about the advantages of the marmoset monkey for the research towards AD at de end of de manuscript under section 5.
Line 297-306: During the 65 million years of evolutionary separation between rodents and primates, neuronal pathways and cognitive capacities have evolved in distinct directions, resulted in specialized brain structures for perceptual and cognitive capacities in primates, including human [114]. Similarities between human and primates include encephalization and sulcal characteristics, comparable numbers and densities of cortical neurons, a large prefrontal cortex containing areas that are responsible for working memory, executive function and aspects of decision-making, similar nuclear organization, projection pathways and innervation pathways of the hippocampus, analogous blood–brain barrier structure and functioning, and the existence of mirror neurons [111].
Line 310-316: More specifically, the marmoset appears to be a good preclinical translational model for immune-related diseases and neuroscience [117]. It also appears that in marmosets, microglia activation persists long after injury and the progression of the microglia response persists, while in rodents it appears in peaks and disappears rapidly after injury [118]. In contrast to rodents, marmosets do also have an comparable analogue of the major histocompatibility complex (MHC) class II in human, the Caja-DR and DQ (from Callithrix jacchus)[119], which is associated with activated microglia with AD lesions in human [120].
Line 318-320: Common marmosets share ~93% sequence identity with the human genome [121,122].
Line 325-328: Marmosets are outbred species raised under normal exposure to antigens, which has allowed their immune systems to develop normally as is the case in humans. This leads to more variation, but also to a more true-to-life representation of the responses to injury.
Line 345-349: Although the use of NHP evokes ethical hurdles that might limit the application for preclinical studies, the small marmoset (300-450 grams) is an attractive model for trans-lational research. The marmoset model covers most of the basic pathogenic mechanisms of the human disease, including protein misfolding and inflammation. Furthermore, only small amounts of candidate drugs is needed in this small NHP model.
7) Something went wrong with the fonts during the editing process. The italics in section 2 has been removed and the text is outlined.
8) In section 2, we have added some words and some sentences to increase the flow of the review. In some cases we also deleted some text that was irrelevant to the review but only caused noise as a result.
Added line 94: In contrast, the role of microglia is widely recognized as a prominent feature in AD [42].
Added line 107: “Next to this, “also the formation of the Aβ protein
Added line 113: RAGE receptor binding[29,45-47], “on the other hand,” results in the secretion of m-CSF…
Added line 116: Besides to the “activation of microglia by the interaction with Aβ protein,”
Added line 119: “Besides the prominent role of M1 microglia in AD,“ the immune reaction…
Deleted line 121: The immune reaction also changes over time. Although the AD immune reaction starts with M1 (and sometimes M2a) activation, it soon changes in a mixed phenotype consisting of M2a and M2c activation in APP/PS1 transgenic mice[51]
Deleted line 122 ……whereas M2c microglia express TGFβ and IL-10 exerting an anti-inflammatory effect[38].
Added line 319: “This also resulted in a” high resemblance...

Round 2
Reviewer 2 Report
Although certain part of this review article has been revised by the authors, some of the reviewer's comments apparently were ignored or not addressed well.
Author Response
Dear reviewer 2, Manuscript ID: vaccines-1115914
We noticed that you still have some concerns about our manuscript entitled “ Preclinical marmoset model for targeting chronic inflammation as a strategy to prevent Alzheimer’s disease” by Philippens and Langermans, written upon invitation for the special issue nonhuman primates in immune modulation and drug discovery.
We sincerely appreciate the positive feedback, thoughtful comments and helpful suggestions for the improvement of the paper.
It was not intended to make you feel that we would ignore some of the comments. We got different responses from reviewers who went in different directions. It was therefore difficult to process all comments in the manuscript. Although yours comments were much more in line with the intention of this manuscript.
As there is not yet much information available about the pros and cons of the marmoset model for AD, we pointed towards the possibilities to develop a NHP model that mimic most of the features of AD in a natural way. As this is a new model a list about the current applications is too preliminary at this moment. Actual, no papers were found in which a treatment has been tested in the amyloid model for AD in the marmoset monkey.
Nevertheless, we have tried to addressed the issues carefully and incorporated the suggestions in our revised manuscript. On top of the changes we have already added in the manuscript, the additional changes we have made in the manuscript are listed in detail below:
Two figures and one table are added.
Figure 1 is about the processing of APP into amyloid-beta.
Figure 2 is about the interaction of amyloid with microglia.
Table 1 provides an overview of specifications of the marmoset monkey compared to rodents and humans.
We refer to Table 1 in line 318: The close evolutionary consistency including the genome comparability and both the face and construct validity of the human aging process and development of age-related diseases, makes NHP an ideal translational model. Different relevant aspects between rodents, marmosets and human are listed in Table 1.
Two sentences have been added to the introduction to make the focus of this manuscript clearer. Line 77: Because this relationship is difficult to investigate in humans, animal models can be an important link in learning more about the cause and effects.
Line 82: This review discusses some of the cellular players in the immune-related aspects of AD in relation to amyloidopathy and how the marmoset monkey can be used as a model to learn more about the role of the immune system in AD.
We add some extra information about the background of the interaction between microglia and amyloid-beta.
Line 114: Microglia bind to Aβ via receptors including class A scavenger receptor A1, CD36, CD14, α6β1 integrin, CD47 and toll like receptors (TLR2, TLR4, TLR6 and TLR9), which is thought to be part of the inflammatory reaction in AD [Heneka et al., 2018]. The binding of Aβ with CD36, TLR4 and TLR6 results in activation of microglia which start to produce proinflammatory cytokines and chemokines [Heneka et al., 2018].
Line 130: As a response to receptor ligation, microglia begin to phagocytose Aβ fibrils. As a result, these fibrils enter the endosomal/lysosomal pathway. Unlike fibrillar Aβ, which is largely resistant to enzymatic degradation, soluble Aβ can be degraded by a variety of extracellular proteases.
We have add this information also in the figure 2.
In section 5, we added extra information about the immune-related advantages of the marmoset monkey.
Line 349: Key components of the antibody response are functionally conserved between lower primates and man [Quint et al. 1990]. The common marmoset may be useful as an in vivo model of immune function, particularly with regard to the role of interleukins [Quint et al., 1990]. Although the immunome proteins identity percentage to human in rhesus monkeys is very high (96.77 %) these proteins still have a strong overlap in evolutionarily more distant marmosets with human of 94.11 % [Plaza et al. 2019]. Also the immunological changes in common marmosets over their life span revealed several similarities to age-related changes in humans [Mietsch et al 2020].
Reviewer 3 Report
The authors have just added a few sentences, ignoring completely all the comments that was suggested earlier
Author Response
Dear reviewer 3, Manuscript ID: vaccines-1115914
We noticed that you still have some concerns about our manuscript entitled “ Preclinical marmoset model for targeting chronic inflammation as a strategy to prevent Alzheimer’s disease” by Philippens and Langermans.
As we tried to explain in the previous letter, this review is focused on the use of the marmoset as a model for drug development against Alzheimer’s disease (AD) and is written upon invitation for the special issue nonhuman primates in immune modulation and drug discovery.
We sincerely appreciate the positive feedback, thoughtful comments and helpful suggestions for the improvement of the paper.
It was not intended to make you feel that we would ignore the comments. But we got different responses from reviewers who went in different directions. It was therefore difficult (and not within the focus of the article) to process all comments in the manuscript. As this manuscript highlights the potential of the marmoset as a model for the immune related aspects of Alzheimer's, we did not focus on all underlying processes of Alzheimer's, but rather some aspects that can be modulated in the marmoset monkey.
We have tried to addressed the issues carefully and incorporated the suggestions in our revised manuscript. On top of the changes we have already added in the manuscript, the additional changes we have made in the manuscript are listed in detail below:
Two figures and one table are added.
Figure 1 is about the processing of APP into amyloid-beta.
Figure 2 is about the interaction of amyloid with microglia.
Table 1 provides an overview of specifications of the marmoset monkey compared to rodents and humans (which was a request from the other reviewer).
We refer to Table 1 in line 318: The close evolutionary consistency including the genome comparability and both the face and construct validity of the human aging process and development of age-related diseases, makes NHP an ideal translational model. Different relevant aspects between rodents, marmosets and human are listed in Table 1.
Two sentences have been added to the introduction to make the focus of this manuscript clearer. Line 77: Because this relationship is difficult to investigate in humans, animal models can be an important link in learning more about the cause and effects.
Line 82: This review discusses some of the cellular players in the immune-related aspects of AD in relation to amyloidopathy and how the marmoset monkey can be used as a model to learn more about the role of the immune system in AD.
We add some extra information about the background of the interaction between microglia and amyloid-beta.
Line 114: Microglia bind to Aβ via receptors including class A scavenger receptor A1, CD36, CD14, α6β1 integrin, CD47 and toll like receptors (TLR2, TLR4, TLR6 and TLR9), which is thought to be part of the inflammatory reaction in AD [Heneka et al., 2018]. The binding of Aβ with CD36, TLR4 and TLR6 results in activation of microglia which start to produce proinflammatory cytokines and chemokines [Heneka et al., 2018].
Line 130: As a response to receptor ligation, microglia begin to phagocytose Aβ fibrils. As a result, these fibrils enter the endosomal/lysosomal pathway. Unlike fibrillar Aβ, which is largely resistant to enzymatic degradation, soluble Aβ can be degraded by a variety of extracellular proteases.
We have add this information also in the figure 2.
In section 5, we added extra information about the immune-related advantages of the marmoset monkey.
Line 349: Key components of the antibody response are functionally conserved between lower primates and man [Quint et al. 1990]. The common marmoset may be useful as an in vivo model of immune function, particularly with regard to the role of interleukins [Quint et al., 1990]. Although the immunome proteins identity percentage to human in rhesus monkeys is very high (96.77 %) these proteins still have a strong overlap in evolutionarily more distant marmosets with human of 94.11 %[Plaza et al. 2019]. Also the immunological changes in common marmosets over their life span revealed several similarities to age-related changes in humans [Mietsch et al 2020].
Round 3
Reviewer 2 Report
In this revised version, my previous comments were well addressed. Also, the newly-added figures make this version much better.
Author Response
Dear reviewer 2, Manuscript ID: vaccines-1115914
We sincerely appreciate the latest positive feedback on the revised manuscript and all the effort you have put into it. It has resulted in very useful comments and helpful suggestions for improving the article.
Again thanks a lot.
Ingrid Philippens
